# Ultrasound-Guided Trans-Uterine Cavity Core Needle Biopsy of Uterine Myometrial Tumors to Differentiate Sarcoma from a Benign Lesion—Description of the Method and Review of the Literature

**DOI:** 10.3390/diagnostics12061348

**Published:** 2022-05-29

**Authors:** Maciej Stukan, Piotr Rutkowski, Jeremy Smadja, Sylvie Bonvalot

**Affiliations:** 1Department of Gynecologic Oncology, Gdynia Oncology Center, Pomeranian Hospitals, ul. Powstania Styczniowego 1, 81-519 Gdynia, Poland; 2Division of Propedeutics of Oncology, Medical University of Gdańsk, ul. Powstania Styczniowego 9B, 81-519 Gdynia, Poland; 3Department of Soft Tissue/Bone Sarcoma and Melanoma, Maria Sklodowska-Curie National Research Institute of Oncology, Roentgena Str. 5, 02-781 Warszawa, Poland; piotr.rutkowski@pib-nio.pl; 4Department of Interventional Radiology, Institut Curie, University of Paris, 26 Rue d’Ulm, CEDEX 05, 75248 Paris, France; jeremy.smadja@curie.fr; 5Department of Surgical Oncology, Institut Curie, University of Paris, 26 Rue d’Ulm, CEDEX 05, 75248 Paris, France; sylvie.bonvalot@curie.fr

**Keywords:** sarcoma, leiomyoma, uterus, ultrasound, core needle biopsy, tru-cut biopsy, preoperative, differential, diagnosis, technique

## Abstract

Uterine sarcomas are rare, aggressive tumors with poor prognosis that can be further negatively affected by inadequate surgical approaches such as morcellation. There are no clinical and radiologic criteria for differentiating leiomyoma from malignant uterine tumors. However, some ultrasonography and magnetic resonance imaging findings may be informative. We present a technique of ultrasound-guided trans-uterine cavity (UG-TUC) core needle biopsy for uterine lesions. As the procedure is an in-organ biopsy, there is no risk of needle canal contamination. The technique also enables the biopsy of lesions inaccessible by the transvaginal tru-cut biopsy. The core needle of the automatic biopsy system is inserted via the cervical canal into the uterine cavity and is directed and activated at the myometrial lesion under ultrasound control. The standard local treatment of localized uterine sarcomas is en bloc total hysterectomy; for fibroids, there are multiple options including conservative management or tumorectomy and tumor morcellation using minimally invasive techniques. Fragmentation of the sarcoma significantly worsens oncologic outcomes and should therefore be avoided. The UG-TUC core needle biopsy of uterine lesions can complement imaging to obtain sufficient material for histologic and molecular analyses of suspected or undetermined lesions, thus facilitating treatment planning and decreasing the risk of unsuspected sarcomas.

## 1. Introduction

Fibroids are common benign uterine tumors affecting approximately 70% of the female population. Treatment is required in 15–30% of female patients [1]. Management is individualized accordingly to age, hormonal status, fertility needs, symptoms, and tumor location. There are various surgical management options including total hysterectomy, supracervical hysterectomy, or myomectomy that are performed via open surgery or minimally invasive techniques. Submucosal fibroids can be managed with hysteroscopic electroresection. Next option is transvaginal hysterectomy. When minimally invasive techniques or a transvaginal approach are used for the treatment of large (presumed) fibroids, the final extraction of the specimen is preceded by the fragmentation of the lesion or whole uterus, which is usually morcellation with the laparoscopic approach. Other nonsurgical treatment options are uterine artery embolization and magnetic resonance imaging (MRI)-guided high-intensity focused ultrasound ablation.

Uterine sarcomas are rare, aggressive tumors with a poor prognosis that can be further negatively affected by inappropriate surgical procedures resulting in intraperitoneal tumor cell spillage such as morcellation outside the laparoscopic bag [2,3,4]. The risk of an unexpected sarcoma is approximately 1:352 [5]. According to national registries in Norway, uterine sarcoma prevalence was 3.6 per 1000 laparoscopic hysterectomies, and the risk of dying from uterine sarcoma after morcellation was 1.5 in 1000 procedures [3]. The correct preoperative diagnosis of these rare tumors is crucial for preventing the inappropriate management of unsuspected sarcomas.

The main imaging modality used for primary soft tissue tumors is MRI [6], which offers an outstanding and comprehensive view of the size, site, and distribution of leiomyomas. However, differentiating between benign degenerating cellular leiomyomas and leiomyosarcomas is challenging because there is considerable overlap in their MRI features [7]. Contrast-enhanced computer tomography (CT) is the standard imaging method for retroperitoneal sarcomas [8]. Ultrasonography may be used as a first-line modality, but if there is any suspicion of soft tissue sarcoma it should be followed by CT or MRI [6]. However, as for uterine tumors, there are no clinical and radiologic criteria to confidently differentiate leiomyoma from malignant uterine tumors [6].

Standard local treatment of localized uterine sarcomas is en bloc total hysterectomy (including laparoscopy/assisted or robotic surgery, provided that the tumor is resected according to the same criteria as for open surgery and morcellation is not carried out) [6]. Morcellation and the spilling of malignant cells alter the natural spread pattern and increase the risk of transperitoneal dissemination and cancer recurrence [9]. The European Society of Gynecological Oncology (ESGO) statement on fibroids and uterine morcellation emphasizes that improvements in preoperative workup are needed to decrease the number of unsuspected sarcomas [5].

Transvaginal core needle biopsy targeting pelvic tumors is an established practice in gynecologic oncology [10,11,12,13]. However, it is not mentioned as a specific recommendation in the preoperative workup for patients with uterine tumors [5]. The guidelines of various sarcoma societies recommend avoiding transabdominal core needle biopsy [6,8]. Moreover, sarcomatous tissue can become embedded in the needle canal [14]. The adequate and safe pretreatment biopsy of lesions suspected to be a sarcoma by imaging is essential and obligatory for appropriate treatment planning [8].

The first aim of this study was to present a technique of ultrasound-guided trans-uterine cavity (UG-TUC) core needle biopsy of uterine myometrial lesions as a procedure complementary to imaging for obtaining adequate specimens for histologic examination prior to treatment. The second aim was to review the literature on the preoperative workup and treatment planning for patients with atypical uterine lesions by imaging.

## 2. The UG-TUC Core Needle Biopsy Technique

The uterus has a cavity and natural orifices and a core needle biopsy of the uterine lesion can be performed via the trans-uterine cavity approach. In this type of in-organ biopsy, there is no risk of core needle canal contamination because the needle only moves within the uterus—i.e., the organ that will be totally resected if a sarcoma is present. The UG-TUC core needle biopsy technique is presented in Figure 1 and Appendix A.

The patient was diagnosed with a tumor localized in the myometrium in the uterine doom (Figure 2 and Appendix A). The patient had an intrauterine device (Figure 2), but this had no relevance to the procedure.

The Appendix A shows an ultrasound examination followed by a hysteroscopic view of the uterine cavity (no lesion nor endometrium deformation was detected). The preparation of the single-use automatic biopsy system (Themy; M.D.L. Srl, Delebio, Italy) with specific adaptation of the core needle cover is shown. The performance of the UG-TUC core needle biopsy of the lesion under transabdominal ultrasound control is presented in Figure 3.

In the single-use automatic biopsy system with the core needle, the plastic needle cover—which is as long as the needle after firing—is intentionally cut at the level, where the needle is set to ready-to-fire mode. It is important to leave the needle cover on the needle—otherwise, the sharp end of the needle would become stuck in the cervical canal, preventing its insertion into the uterine cavity. Once the cover is shortened, the gun is set to the ready-to-fire mode. The shortened needle cover is left on the needle to allow safe and easy insertion of the system through the uterine cervix canal into the uterine cavity. The patient is in a gynecologic position. The procedure is performed under short intravenous anesthesia (the same as that provided for uterine curettage or hysteroscopy). A speculum is inserted into the vagina and the uterine cervix is visualized. Diagnostic hysteroscopy is first performed to visualize the uterine cavity and the position of the uterine corpus (the smooth endometrium and intrauterine device are also visible in the video). The hysteroscope and intrauterine device are withdrawn; a convex probe is placed over the symphysis pubis in the longitudinal plane, tilted in the caudal direction, and the uterus is visualized (in the longitudinal plane). The ultrasound device is operated by a second physician (not the one performing the biopsy). Transrectal ultrasonography can also be performed (Figure 1).

The automatic biopsy system (with the covered needle) is inserted through the cervical canal into the uterine cavity. The position of the needle is visualized by ultrasonography and the needle is directed toward the uterine tumor. The automatic biopsy system is activated and ultrasound-guided core biopsy is performed. The system is withdrawn from the uterus and the specimen is obtained from the needle. The procedure is repeated three or more times, directing and activating the needle at a different angle at the lesion each time to obtain a more representative specimen. The procedure can be followed by a biopsy of the endometrium and cervical canal mucosa, either by curettage or using hysteroscopic tools.

## 3. Discussion

Minimally invasive surgical techniques are widely used in gynecology. The radicality of procedures is limited whenever possible, especially when managing benign lesions such as leiomyomas. On the other hand, the standard local treatment of localized uterine sarcomas is en bloc total hysterectomy [6], and procedures that can potentially cause tumor cell spillage out of endoscopic bags such as morcellation are discouraged because they can worsen patient prognosis when the postoperative pathologic diagnosis is malignancy [6]. A meta-analysis showed that morcellation increased overall (62% vs. 39%; odds ratio [OR] = 3.16, 95% confidence interval [CI]: 1.38, 7.26) and intra-abdominal (39% vs. 9%; OR = 4.11, 95% CI: 1.92, 8.81) recurrence rates as well as mortality rates (48% vs. 29%; OR = 2.42, 95% CI: 1.19, 4.92) [15]. In 1367 patients with uterine sarcoma in a national registry from Norway, sarcoma mortality was higher in the morcellated group than in the non-morcellated group (age-adjusted hazard ratio [HR] = 1.90, CI: 1.05, 3.44; multivariate HR = 2.50, 95% CI: 0.57, 10.9); and age-adjusted 10-year uterine sarcoma survival was 32.2% vs. 57.2% (difference of 25.5%; CI: −55.7, 18.1) [3]. In a retrospective multicenter analysis of 125 patients with uterine sarcomas, those who underwent morcellation had a threefold higher risk of death compared to patients who did not (*p* = 0.02). A trend toward the increased rate of recurrence was observed for patients who underwent morcellation for smooth muscle tumors of uncertain malignant potential (HR = 7.7, *p* = 0.09); whereas no differences in survival outcomes were observed for patients with low-grade endometrial stromal sarcomas and undifferentiated uterine sarcomas [2]. In another retrospective cohort study conducted at a sarcoma referral center, intraperitoneal morcellation was associated with a significantly increased risk of abdominal/pelvic recurrences (*p* = 0.001) and significantly shorter median recurrence-free survival (10.8 vs. 39.6 months; *p* = 0.002). A multivariate adjusted model demonstrated a >three times increased risk of recurrence associated with morcellation (HR = 3.18, 95% CI: 1.5, −6.8; *p* = 0.003) [16]. Among women with occult uterine sarcomas, laparoscopic supracervical hysterectomy/laparoscopic myomectomy was associated with a higher risk of disease-specific mortality than total abdominal hysterectomy (adjusted HR = 2.66, 95% CI: 1.11, 6.37; adjusted difference in 5-year disease-specific survival of −19.4%, 95% CI: −35.8%, −3.1%). Differences were even more significant in the subset of women with leiomyosarcoma [17].

The decision regarding the optimal surgical treatment of uterine tumors—i.e., minimally invasive excision of the tumor and its morcellation or other methods of fragmentation, or supracervical or total hysterectomy (which also applies to transvaginal hysterectomy for presumed uterine fibroids, for which fragmentation of the enlarged uterus is frequently performed)—is irreversible. There is no issue in the case of benign tumors, but if sarcoma is postoperatively diagnosed, the surgical outcome cannot be reversed or corrected.

At the reexploration of patients with incidentally diagnosed uterine malignancy following morcellation or supracervical hysterectomy for presumed benign uterine disease, 15–28% of patients were upstaged [18,19]. Extensive secondary surgery for incidentally morcellated uterine sarcomas does not always yield the expected oncologic outcome; in a case series, only 1 of 5 patients had longer survival after cytoreductive surgery and hyperthermic intraperitoneal chemotherapy for morcellated uterine sarcoma [20]. The application of adjuvant systemic treatment also did not improve outcomes for patients after sarcoma morcellation: gemcitabine-docetaxel had no effect [21], and no difference was observed between adjuvant anthracycline-based vs. gemcitabine-based chemotherapy [22].

An electric device for the morcellation of uterine lesions was found to be suitable for a minimally invasive approach, and many patients can benefit from this tool. However, it must be used for the appropriate patient in the proper way. Tumorectomy and morcellation of uterine lesions using the spill-free technique described by Haak et al. could be recommended to maximize the safety of surgery for individuals who request a minimally invasive approach [23]. Preoperative protocols such as routine endometrial sampling, laboratory analyses, and imaging are also recommended to detect occult pathology prior to surgery [4].

There are no clinical or radiologic criteria to confidently differentiate leiomyoma from malignant uterine tumors [6]. Several features on ultrasonography and MRI can raise suspicion of uterine sarcoma, but neither method is 100% accurate [9]. Most uterine leiomyosarcomas have a large oval shape with a nonhomogeneous and bizarre internal echo pattern on ultrasonography, with mixed echogenic and poorly echogenic areas surrounded by a thinned myometrium. Central necrosis is common. Findings on color Doppler ultrasonography include irregular vessel distribution within the tumor, with low impedance to flow (as reflected by low resistance indices) and high peak systolic velocities. However, there is considerable overlap between these findings and those of benign leiomyomata [24,25]. On ultrasonography, endometrial stromal sarcomas can present as a hypoechogenic mass with irregular margins originating from the endometrium and irregular central or circular vascularization. A heterogeneous pattern of the endometrium with high-intensity and hypoechoic areas scattered in the myometrium has also been linked to endometrial stromal sarcomas [25,26]. Undifferentiated endometrial sarcomas often have no shadowing, irregular tumor borders, and hemorrhagic or ground-glass echogenicity of cyst fluid [27]. Importantly, when ultrasonography was performed by an experienced sonographer, >80% of uterine sarcomas were described as atypical lesions and 78.5% were suspected of malignancy [27].

Four discriminative MRI features are nodular borders, hemorrhage, dark areas on T2-weighted imaging, and central unenhanced areas; combining ≥three MRI features yielded a specificity >95% [28]. Scattered hemorrhagic or necrotic foci are suggestive of uterine leiomyosarcoma; these are visible as areas of slightly higher intensity on T1-weighted images and as heterogeneous areas on T2-weighted images. A consistent finding in uterine leiomyosarcomas is the absence of calcifications [25,29,30]. On MRI, endometrial stromal sarcomas typically present as an invasive endometrial mass with extensive myometrial involvement that is either sharply demarcated or diffusely infiltrative. On T2-weighted images, bands of low signal intensity corresponding to preserved bundles of myometrium are noted within areas of myometrial involvement. Another common MRI finding in endometrial stromal sarcomas is tumor extension along the vessels or ligaments [25,29,31].

It has been suggested that the accuracy of uterine sarcoma detection may be improved by the simultaneous use of multiple diagnostic modalities. An example is the combined use of MRI (including dynamic MRI) and serum lactate dehydrogenase levels [32]; another is the addition of 16α-[^18^F]fluoro-17β-estradiol positron emission tomography (PET) to [^18^F]fluorodeoxyglucose PET [33]. Both approaches had higher accuracy than the individual methods; however, they were investigated in single-institution studies in a relatively small number of patients and were not validated.

The ESGO statement on fibroids and uterine morcellation recommends that ultrasound examination be performed by an experienced sonographer in patients for whom myomectomy or hysterectomy with morcellation is being considered. Additionally, when power morcellation is planned, a preoperative endometrial biopsy with hysteroscopy is mandatory [5]. However, it was reported that leiomyosarcomas were diagnosed in just 35% and endometrial stromal sarcomas in 25% of patients undergoing endometrial biopsy [34]. By definition, an endometrial biopsy does not target myometrium lesions. In contrast to the abovementioned proposals for improving the pretreatment workup, we suggest that an image-guided, targeted biopsy be performed if possible for every atypical uterine lesion via a transuterine (in-organ) approach.

The core needle biopsy of pelvic tumors is well established [10,11,12,13]. The accuracy of the biopsy diagnosis in 118 patients who underwent ultrasound-guided core needle biopsy with an 18-gauge needle at a gynecology clinic was 98.3% (no patients had sarcoma) [10]. In 62 gynecologic patients, the final diagnosis based on histology was not in agreement with transvaginal ultrasound-guided biopsy results in two patients (20%), both of whom had a recurrence of cervical cancer (there were no patients with sarcoma) [13]. In another study, the overall diagnostic accuracy of ultrasound-guided transvaginal core needle biopsy was 93%; of the four patients with discordance between biopsy and final histology results, one had leiomyosarcoma [12].

In a meta-analysis of 32 studies comprising 7209 musculoskeletal lesions, the concordance between core needle biopsy and final histology results was 84% [35]. For retroperitoneal leiomyosarcomas, the concordance between results obtained by percutaneous core needle biopsy and definitive resection was as high as 81.1% [36].

The open or laparoscopic biopsy of lesions suspected of being sarcomas must be avoided [6], as it could open the tumor into the abdominal cavity. Using a frozen section from the laparoscopic biopsy of a uterine tumor before its morcellation for diagnosis has been suggested [37], but this approach would risk the contamination of the abdominal cavity if sarcoma is present. Moreover, it does not allow a complete diagnosis of sarcoma [6,8], as a frozen section may underestimate the tumor grade [6] and prohibit complex immunohistochemistry or other molecular analyses that may be required. If a retroperitoneal tumor is found incidentally during surgery for a presumed ovarian or uterine tumor, definitive surgery should not be attempted and core needle biopsy should be considered. The patient should also undergo subsequent dedicated imaging and treatment planning that should be carried out by a multidisciplinary tumor board at a center specializing in the treatment of sarcoma [8].

It is unclear why the core needle biopsy is still not the standard of care or even recommended before treatment decisions are made for uterine tumors that are atypical or undetermined in imaging. Obtaining an adequate tissue sample is critical during a pretreatment workup, as it enables histologic evaluation of the specimen. When there is clinical suspicion of sarcoma, the biopsy specimen can be sent to a sarcoma reference center for detailed examination. In addition to immunochemistry, more sophisticated methods can be applied. In uterine tumors with uncertain malignant potential, the assessment of genomic index by comparative genomic hybridization array, that is, counting the genomic complexity of a tumor, allows leiomyosarcomas to be distinguished from benign tumors such as leiomyomas [38].

The recurrence of sarcoma in the needle tract is rare but possible [14]. Recent guidelines from various sarcoma societies state that the biopsy should be planned in such a way that the biopsy tract and scar can be safely removed during the definitive surgery; the biopsy entrance point can even be tattooed [6].

The previously described ultrasound-guided core needle biopsy of pelvic tumors was performed transvaginally [10,11,12,13]—that is, the needle crossed the vaginal wall, paracolpium, retroperitoneal tissues around the uterus, and the abdominal cavity some of the time. When uterine sarcoma is diagnosed, needle tract resection during definitive surgery may not be possible because of anatomic constraints. Moreover, when using the transvaginal ultrasound probe with the needle guide fixed to the probe, clinicians focus on targeting the pelvic lesion, but it would be not feasible to mark the biopsy entrance. The vaginal fornix could be resected along with the uterus, but most clinicians would rather perform a simple hysterectomy for sarcoma while leaving the vaginal fornix intact. Moreover, a core needle biopsy performed via the transvaginal approach can pass through another part of the vaginal wall besides the fornix. Another issue is that the needle can penetrate the paracolpium and retroperitoneal space around the uterus, areas that would probably not be resected during the surgery. The worst outcome of transvaginal core needle biopsy is if the needle penetrated the abdominal cavity (e.g., from the posterior vaginal fornix through the pouch of Douglas to the uterine sarcoma). In a set of 255 patients with retroperitoneal sarcomas diagnosed based on a preoperative biopsy, all five biopsy site recurrences occurred after transabdominal biopsies that were not performed using a coaxial technique [14].

The transvaginal core needle biopsy procedure is generally safe; reported complications include vaginal bleeding (18–100%) and hematuria (4%), and none of these cases required any active management [12,13]. One study reported that two of 195 patients (1%) who underwent transvaginal or transabdominal biopsy experienced bleeding into the peritoneal cavity that required laparotomy. One of these patients suffered thrombocytopenia and bleeding from the ovarian metastatic tumor, and the other had bleeding from the site of the pelvic carcinomatosis biopsy (that eventually stopped spontaneously) [10]. With the exception of uterine bleeding, the abovementioned complications are unlikely if the biopsy of uterine lesions is performed via the UG-TUC approach.

The main advantage of the UG-TUC approach is that the needle tract is within the uterus only, which prevents any cell spillage (which would be harmful if sarcoma is diagnosed). Another is that the procedure is to some extent similar to curettage or hysteroscopy, and can therefore be performed by most gynecologists, the difference being that ultrasound is used to guide the needle. Additionally, after the UG-TUC core needle biopsy, endometrium and cervical canal mucosa biopsy specimens can be obtained.

Gynecologists typically use 18-gauge core needles to perform a transvaginal biopsy of pelvic lesions [10,11,12,13], which is likely dictated by the available needle guide for transvaginal probes. However, multidisciplinary clinical guidelines recommend the use of 14- or 16-gauge needles [6,8]. Specimens collected with a smaller diameter needle (e.g., 18-gauge) may be of lower quality for histologic and molecular analyses and may thus yield less accurate results [8]. With the UG-TUC approach, the needle guide is not required and needles larger than 18 gauge can therefore be used.

A limitation of the UG-TUC core needle biopsy method is that many different angles for the needle tract are not possible because the biopsy system is “fixed” by the cervical canal. A greater range of motion can be achieved by pressing on the uterine corpus from the anterior abdominal wall, even with the ultrasound probe, with simultaneous visualization of the needle. However, this maneuver would be impossible in patients who are obese or do not have an enlarged uterus. Additionally, there are lesions that cannot be targeted with the UG-TUC approach. Finally, a caveat of using the UG-TUC approach is that its accuracy and safety and the subsequent pathologic analysis have not been extensively validated because of the rarity of uterine sarcomas. However, a case series is provided in Appendix B, where we present the clinical usefulness and value of the core needle biopsy as an additional diagnostic tool for lesions of the myometrium that are atypical in ultrasound or clinically.

## 4. Conclusions

Preoperative core needle biopsy of uterine tumors that are atypical or undetermined by ultrasonography should be considered whenever possible as an adjunctive tool to imaging, as it allows the possibility of histologic differentiation between atypical uterine leiomyoma or sarcoma prior to management decisions; oncology patients can then be referred to a specialized center for surgery. It would also allow for the determination of benign lesions and the safe management of patients who opt for a minimally invasive or transvaginal approach or a nonsurgical treatment.

Preoperative core needle biopsy of uterine tumors should be more widely used and available. Both transvaginal and UG-TUC approaches can be adopted, whichever is more applicable to the individual patient and tumor location, with greater consideration given to the latter because of its oncologic cleanness (in-organ biopsy). Future research should focus on further improving the accuracy of the preoperative workup, which could include the use of radiomics for imaging and the addition of core needle biopsy of uterine lesions including the UG-TUC approach. Multicenter, prospective studies with standardized imaging terms and definitions [39] and biopsy procedures are needed to establish the feasibility, effectiveness, safety, and clinical utility of core needle biopsy for uterine sarcomas. Given the rarity of this entity, the research should be carried out by an international network for rare tumors.

## Figures and Tables

**Figure 1 diagnostics-12-01348-f001:**
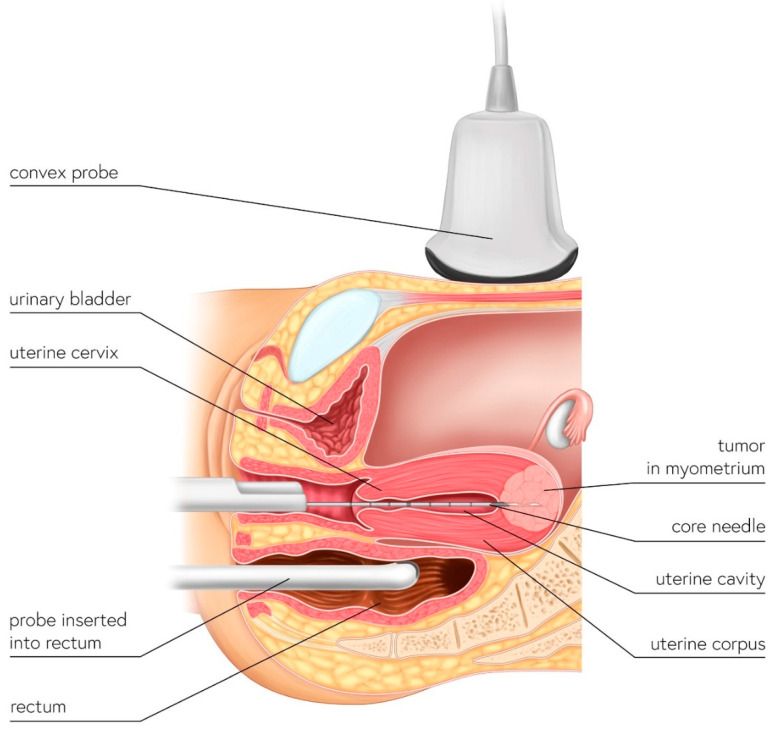
Schematic illustration of UG-TUC core needle biopsy of myometrial lesions. The biopsy needle is inserted into the uterine cavity (in-organ biopsy) and the lesion is targeted by ultrasonography. Either a transabdominal (with a convex probe) or transrectal (e.g., even with a transvaginal probe) approach—whichever provides better visualization—can be used.

**Figure 2 diagnostics-12-01348-f002:**
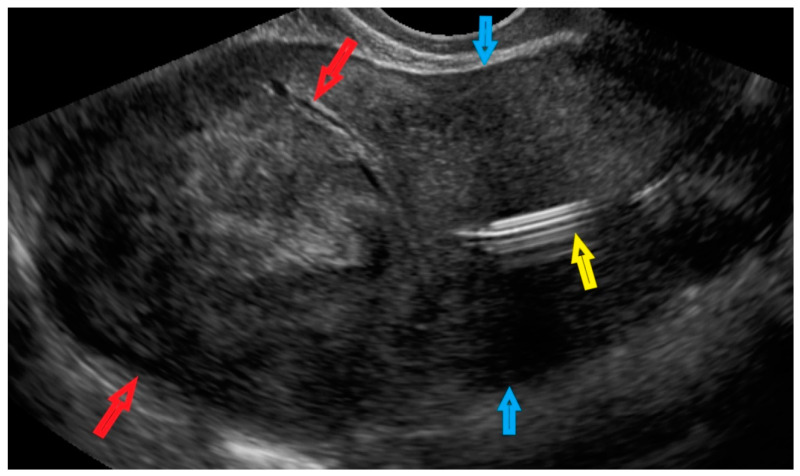
Transvaginal ultrasonogram of the uterus in a patient. Red arrows indicate a lesion in the myometrium in the uterine doom; blue arrows indicate the uterine corpus; and the yellow arrow indicates an intrauterine device.

**Figure 3 diagnostics-12-01348-f003:**
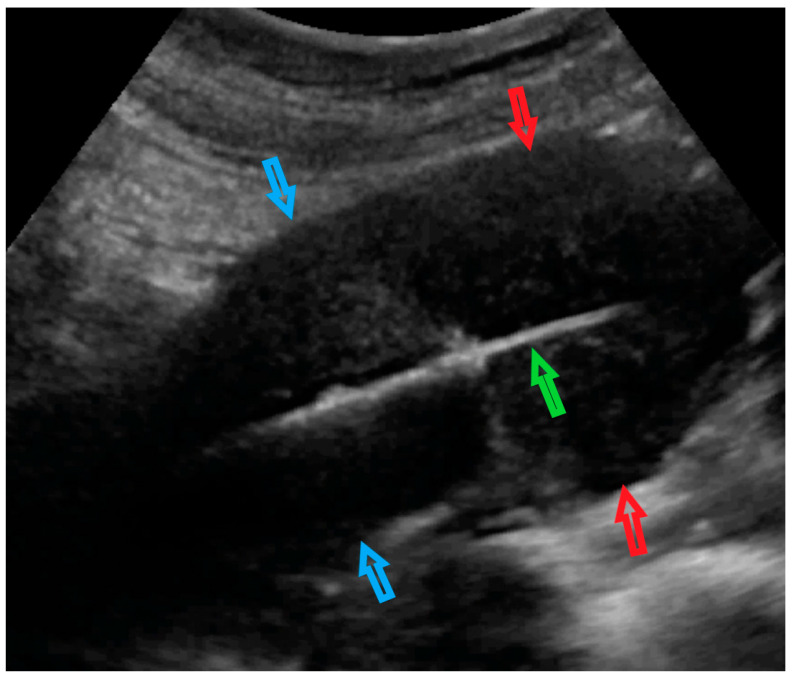
Transabdominal ultrasonogram of the uterus and core needle biopsy of the uterine lesion. The red arrows indicate the uterine lesion (in the uterine doom myometrium); blue arrows indicate the uterine corpus; the green arrow indicates the automatic biopsy system in the uterine cavity. The core needle is activated into the uterine myometrium lesion.

## Data Availability

Not applicable.

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
