# Peer review of "Ultrasound-Guided Trans-Uterine Cavity Core Needle Biopsy of Uterine Myometrial Tumors to Differentiate Sarcoma from a Benign Lesion—Description of the Method and Review of the Literature"

_diagnostics, 2022, doi:10.3390/diagnostics12061348_

Round 1

Reviewer 1 Report

I read your manuscript with great interest and agree with your conclusion that ultrasound-guided trans-uterine cavity (UG-TUC) core needle biopsy would certainly be beneficial to the patients who turns out to have uterine sarcomas.  However, as you pointed out in your manuscript, it needs to be utilized in select patient population where the imaging findings raise the possibility of uterine sarcoma.  

Author Response

Dear Reviewer,

Thank you very much for your opinion and valuable comment on our article. We very much appreciate your experience in the field, and that you had time and took the effort to review the manuscript.

Thank you for pointing out the issue that the UG-TUC core needle biopsy can be utilized in a selected patient population (not all), where the imaging findings raise the possibility of uterine sarcoma. It is expressed in the conclusions section.

The expert ultrasonography assessment of uterine lesions is the first-line and most often the only needed diagnostic tool to plan clinical management. However, atypical lesions are to be expected, and then the core needle biopsy should be considered.

After the review, we added Appendix A, where we presented the case series with the application of the core needle biopsy as an additional tool to ultrasonography. Most of the myometrium lesions were considered atypical in ultrasound. In some cases, the core needle biopsy results helped to make the optimal decision about clinical management.

Yours sincerely,

in the name of all authors,

Maciej Stukan

Reviewer 2 Report

The article presents an interesting line of work with the contribution of diagnostic information in the search for pathological cases that can penalize the prognosis of patients. The authors' proposal is good and the reflection in the use of a new device seems correct to us.
However, every mechanism or instrument in its clinical application must be supported by its results. The article details the cases in which it should be used and the mechanism of operation and use of it. But it would reinforce and validate its use if the authors, even if they were, presented some of their results. The design can be very successful but it must be backed by a clear utility and this is inexorably backed by the results, complication rate, false negatives, etc.

Author Response

Dear Reviewer,

Thank you very much for your opinion and valuable comments on our article. We very much appreciate your experience in the field, and that you had time and took the effort to review the manuscript.

Thank you for pointing out the issue that the clinical application of the UG-TUC core needle biopsy must be supported by its results. We acknowledge this and expressed it at the end of the manuscript (lines 333-338, the revised version), where we discuss limitations of the our project. Also, we addressed this issue in the conclusions section (lines 354-358, the revised version). Given the rarity of uterine sarcomas, the research on the feasibility, accuracy, and safety of the UG-TUC core needle biopsy should be carried out by many clinicians in an international network for rare tumors.

Encouraged by your suggestion (“the authors, …, presented some of their results”) we added the Appendix A, where we presented the case series with the application of the core needle biopsy as an additional tool to ultrasonography. Most of the myometrium lesions were considered atypical in ultrasound. In some cases, the core needle biopsy results helped to make the optimal decision about clinical management.

We consider Appendix A as an added value to the manuscript, thank you for the suggestion, and hope you will accept our response.

Yours sincerely,

in the name of all authors,

Maciej Stukan

Reviewer 3 Report

This review paper, entitled Ultrasound-guided trans-uterine cavity core needle biopsy of uterine myometrial tumors to differentiate sarcoma from a benign lesion – description of the method and review of the literature, was very excellent in describing advantages, disadvantages and complications. The authors also showed the method detailed using the video. It is helpful in clinical practice. I suggest accept in present form.

Author Response

Dear Reviewer,

Thank you very much for your opinion on our article. We very much appreciate your experience in the field, and that you had time and took the effort to review the manuscript.

We thank you for your acclaim of the manuscript in the presented form. We are very happy about it.

Yours sincerely,

in the name of all authors,

Maciej Stukan